# Narrow-Spectrum Antibacterial Agents—Benefits and Challenges

**DOI:** 10.3390/antibiotics9070418

**Published:** 2020-07-17

**Authors:** Richard A. Alm, Sushmita D. Lahiri

**Affiliations:** 1CARB-X, Boston University, Boston, MA 02215, USA; 2Boehringer-Ingelheim, Cambridge, MA 02142, USA; sushmita.lahiri@boehringer-ingelheim.com

**Keywords:** antibacterial, nontraditional, narrow-spectrum, research and development, diagnostics

## Abstract

The number of antibacterial agents in clinical and preclinical development possessing activity against a narrow spectrum of bacterial pathogens is increasing, with many of them being nontraditional products. The key value proposition hinges on sparing antibiotic use and curtailing the emergence of resistance, as well as preventing the destruction of a beneficial microbiome, versus the immediate need for effective treatment of an active infection with a high risk of mortality. The clinical use of a targeted spectrum agent, most likely in combination with a rapid and robust diagnostic test, is a commendable goal with significant healthcare benefits if executed correctly. However, the path to achieving this will come with several challenges, and many scientific and clinical development disciplines will need to align their efforts to successfully change the treatment paradigm.

Appropriately, the world’s infectious disease focus has abruptly turned to the global COVID-19 response; however, the crisis of antimicrobial-resistant bacterial infections and the associated risk on global health should not be ignored. Many articles have highlighted the health and economic impact of the increasing levels of infections caused by drug-resistant bacteria coupled with the declining number of products being clinically developed and large companies exiting the area [1,2,3]. Further, emerging literature of the management of COVID-19 patients shows that almost 3 in 4 received empirical antimicrobial treatment despite only a relatively small number having underlying bacterial co-infections [4,5,6], a situation that may exacerbate the underlying antimicrobial resistance threat in hospitals.

For many years, both clinical and commercial factors have driven the use of broad-spectrum agents. The ability to treat infections quickly without waiting for a day or more for the identification of the disease-causing pathogen resulted in saving countless lives. In the pharmaceutical industry, the lure of a broad-spectrum antibacterial drug that could be used empirically and generate a significant revenue stream was paramount. The advent of high-throughput screening for inhibitors on isolated enzyme targets afforded the possibility to recombinantly produce and screen homologous enzymes from a variety of key pathogens that represented the spectrum of pathogens most commonly associated with specific clinical indications. The result of this was that many small molecule inhibitors that possessed a narrow spectrum of activity were actively discarded from lead identification and lead optimization programs in favor of options with broad spectrum activity with higher potential commercial value.

## 1. Why Narrow Spectrum Antibacterial Approaches Are Needed

Physicians have long been aware of the collateral damage that broad-spectrum antibacterial drugs have on the intestinal microbiome and the subsequent risks of adverse events. The most serious of these is diarrhea caused by overgrowth of *Clostridioides difficile* (CDAD), as it replaces the natural microbial flora. While the effects can be mild and self-limited, progression to pseudomembranous colitis can be severe and often fatal, with reported mortality rates in the United States at 12,800 in 2017 [7]. The impact of widespread use of broader-spectrum agents on the human microflora should not be underestimated.

Aside from the adverse events and risk of CDAD noted above is the impact of broad-spectrum agents supporting the emergence of resistance. Resistance can occur directly where the pathogen develops chromosomal-based mutational resistance during treatment. However, there are other mechanisms that impact resistance emergence. Horizontal transmission of resistance elements can come from nonpathogenic environmental species, such as *Shewanella algae*, a marine and freshwater species, from which the plasmid-encoded *qnrA* genes conferring fluoroquinolone resistance originated [8]. Some drugs may have weak activity against bacterial species for which they are not prescribed, but that subinhibitory exposure may result in resistance mechanisms accumulating in those species not considered clinically susceptible to the drug for approved indications. One example of this is the by-product of the widespread use of macrolides to treat upper respiratory tract infections. The oral administration of macrolides exposes the gut flora to these drugs, typically at subinhibitory levels. Despite not being recommended for treatment of Gram-negative infections, *Escherichia coli* isolates have developed a significant pool of preexisting macrolide-specific resistance mechanisms [9]. Sequence analysis of a panel of global contemporary *E. coli* isolates identified macrolide phosphotransferases in approximately 25% of isolates, along with fewer methylase resistance mechanisms [9]. These enzymes cluster with other antibiotic resistance genes on a variety of plasmids, suggesting that further rapid spread by horizontal transmission is possible. A second example of subtle selective pressure involves chromosomal missense mutations in the penicillin-binding protein 2a (PBP2a) of methicillin-resistant *Staphylococcus aureus* (MRSA). A study of PBP2a sequence variants from diverse MRSA isolates identified several that resulted in a decreased susceptibility to ceftaroline, an anti-MRSA cephalosporin agent [10]. Importantly, however, some of these sequence variants were identified in MRSA isolates that had been archived as early as 1998, significantly earlier that the commercial launch of ceftaroline in 2010 [11]. Ceftaroline nonsusceptibility is a complex phenotype [12], and it is hypothesized that biological pressure other than from ceftaroline but likely from other β-lactam agents has led to several independently generated circulating MRSA clones with decreased ceftaroline susceptibility [10,11,13].

Today, essentially all classes of antibiotics are becoming ineffective at a speed faster than the rate of development of novel agents to maintain clinical effectiveness in the face of emerging resistance, and the constant evolution of microbial pathogens able to resist antibiotic treatments is seen as one of the most important public health emergencies. Although making incremental changes to existing chemical scaffolds to improve potency or circumvent an existing resistance mechanism can provide molecules with clinical utility, the risk remains as to how long they will remain useful given the evolution of resistance. New agents are certainly not devoid from resistance emergence; however, the hope is they may slow it down and that, together with the concept of narrow spectrum agents paired with rapid diagnostics coupled with strong stewardship and infection control programs, they may blunt the global threat of antimicrobial resistance.

## 2. Current Narrow Spectrum Antibacterial Efforts

Narrow spectrum agents that act against specific species and do not generate resistance in other pathogens due to selection pressure have become an attractive alternative to combat resistance development while developing effective novel antibacterial drugs. The release of the initial CDC Antibiotic Threats list in 2013 [14]—updated in 2019 [7]—as well as the World Health Organization (WHO) priority pathogen list in 2017 [15] helped focus researchers against the specific pathogens of greatest unmet medical need. This, coupled with the concepts of personalized medicine and improving antibiotic stewardship, has increased the attention on narrow-spectrum or species-specific antibacterial agents. Analysis of the most recent WHO clinical pipeline analyses [16] (and including 2 recent Phase I trial initiations) has identified 17 products (7 small molecule and 10 biologics) being developed for narrow-spectrum use against the WHO priority pathogens (Table 1), which represents approximately half of the current clinical pipeline. More than half of these narrow-spectrum agents are directed against *Staphylococcus aureus*, including 3 small molecules and 6 biological agents, of which 2 are phage lysins and 4 are antibody-based products. In addition, there are 8 agents (6 small molecule and 2 biologics) being developed to treat *C. difficile* infections, and 12 agents in development against *Mycobacterium tuberculosis* and nontuberculosis *Mycobacterium* spp. [16]. The trend toward the discovery of narrow-spectrum agents also continues into the preclinical area. Recently, WHO conducted a global analysis of publically available information on antibacterial development and identified 252 programs affiliated with 145 different institutions [17]. Of these programs, there were 100 (40%) focused against a single pathogen, with the majority targeting *Mycobacterium tuberculosis* (Table 2). Notably, a significant number of programs with a narrow-spectrum focus were ‘’nontraditional products”, including 23 bacteriophage products and 12 antivirulence approaches. Another assessment based on five confidential databases that contained information on preclinical antibacterial research identified 22% of 407 antibacterial projects (*n* = 90) that were focused against a single species [18]. This shift in modality away from direct-acting small molecules toward some of the nontraditional approaches has increased the number of narrow-spectrum agents. This has been unwillingly further supported by the lack of large chemical libraries containing compounds with suitable physicochemical properties for antibacterial potency that provide the diversity needed to identify novel scaffolds and chemical starting points. The exit of large pharma companies and their chemical libraries from this therapeutic area has further limited the discovery of chemical scaffolds that can be active against multiple pathogens. Taken together, this information suggests an unprecedented shift away from broad-spectrum agents and toward narrow-spectrum products and precision medicine, but with this shift will come additional challenges with clinical development, regulatory approval, and ultimately successful clinical uptake.

## 3. Challenges Remain to Develop Narrow Spectrum Antibacterial Agents

While several clinical syndromes are caused by individual bacterial species, the majority can be caused by multiple species, and indeed in some cases can be polymicrobial, with the added challenge of distinguishing pathogen from colonizer. Rapid initiation of appropriate treatment correlates with reduced mortality and improved clinical outcomes, a trend most notable with bloodstream infections and sepsis [19,20,21,22]. This becomes especially important when considering narrow-spectrum agents, as there is a need for rapid identification of the infecting pathogen, along with the susceptibility profile, to ensure the administration of an appropriate agent in a timely fashion. The availability and routine deployment in clinical settings of rapid, sensitive, and easily interpretable diagnostics that support bacterial identification and associated susceptibility information will almost certainly be important to the successful development and clinical uptake of narrow-spectrum agents, especially if considered for empirical use. The field of diagnostic testing continues to evolve rapidly, and new molecular technologies are able to provide clinicians with information significantly faster [23]. The continual development and improved access and adoption of these technologies to support clinical decision-making will benefit the patient, especially in acute infection syndromes. The alternative role would be to reserve these narrow-spectrum agents for combination therapy or used during de-escalation after the causative pathogen has been determined. If this de-escalation takes several days for routine culture and susceptibility results, the overall goal of improved stewardship practices and minimizing resistance emergence will not be fully realized due to the continued reliance of first-line broad spectrum agents in these scenarios.

Challenges also exist in the path to registration for some of these narrow-spectrum agents, many of which are also nontraditional, as they will still be required to meet the exacting standards of demonstrating clinical benefit within the current regulatory guidelines. Clinical trial designs need to identify and enroll suitable patients where this clinical benefit must be clearly demonstrated. The prevalence of certain pathogens will likely be a factor, with less common species leading to a longer and more costly development path. Traditional direct-acting small molecules that possess a narrow spectrum of activity, for example, afabicin targeting the *Staphylococcal* FabI protein, can follow a more standard development path to demonstrate clinical benefit using standard noninferiority trial designs with standard-of-care comparators. Bacteriophage therapy is an area of increasing preclinical (Table 2) and early clinical (Table 1) activity. Bacteriophages are inherently narrow spectrum, and this enables precise targeting of specific species. While there have been clear examples of clinical benefit afforded to individual patients with personalized bacteriophage therapy administered under emergency IND authorizations [24,25,26], several groups are now moving forward in a more traditional development path that includes adequate and well-controlled clinical trials with both wild-type and CRISPR-engineered bacteriophage products (Table 1). Unlike these direct-acting approaches, the-nontraditional narrow spectrum agents that target a virulence mechanism that aim to reduce the pathogenicity of the organism will likely be used as adjunctive therapy in combination to improve the effect of standard antibacterial agents. There are immediate challenges of determining a suitable human dose or developing a clinical microbiology test to measure efficacy and monitor resistance emergence of a product that does no inhibit cellular growth. However, beyond these challenges will be one of clinical trial design. The requirement to augment the clinical benefit of an underlying agent will necessitate clinical trials designed to demonstrate statistical superiority over using the underlying agent alone, which, given the efficacy of most standard-of-care antibacterial agents, will be a challenging task.

In summary, there are clear clinical and societal benefits for treatment paradigms employing rapid diagnosis and use of targeted narrow-spectrum agents. These include improved antibiotic stewardship, reduced levels of emerging resistance, less collateral damage on the microbiome, and enhanced personalized medicine. There is an increasing effort to develop narrow spectrum agents, as can be seen in the current pipeline and the growing number of narrow-spectrum and nontraditional agents being studied in response to the growth threat of widespread antibacterial resistance cannot be ignored. The learnings from the clinical development efforts of these products, together with the growing landscape of rapid diagnostics, need to be efficiently leveraged. The road to success in developing these products will be challenging, but often the most worthwhile journeys are.

## Figures and Tables

**Table 1 antibiotics-09-00418-t001:** Narrow-spectrum agents in clinical development.

Product Name (Synonym)	Product Class (Description)	Target Species ^a^	Development Stage	Clinical Trial Identifier	Developer
Durlobactam + sulbactam (ETX2514SUL)	Small molecule ^b^	*Acinetobacter baumannii*	Phase III	NCT02971423, NCT03445195	Entasis Therapeutics
Zoliflodacin (ETX0914)	Small molecule	*Neisseria gonorrhoeae*	Phase III	NCT02257918, NCT03959527	Entasis Therapeutics/GARDP
Afabicin (Debio-1450)	Small molecule	*Staphylococcus aureus*	Phase II	NCT02426918	Debiopharm International
AR-501 (Panaecin)	Small molecule	*Pseudomonas aeruginosa*	Phase I	NCT03669614	Aridis Pharmaceuticals
TXA709	Small molecule	*Staphylococcus aureus*	Phase I	Not registered	Taxis
TNP-2198	Small molecule	*Helicobacter pylori*	Phase I	Not registered	TenNor Therapeutics
BCM-0184	Small molecule	*Staphylococcus aureus*	Phase I	Not registered	Biocidium Biopharmaceuticals
AR-301 (Salvecin)	Biological (monoclonal Ab)	*Staphylococcus aureus*	Phase III	NCT03816956	Aridis Pharmaceuticals
CF-301 (Exebacase)	Biological (phage endolysin)	*Staphylococcus aureus*	Phase III	NCT03163446, NCT03446053	Contrafect
SAL-200 (tonabacase)	Biological (phage endolysin)	*Staphylococcus aureus*	Phase II	NCT03089697, NCT03446053	Intron Biotechnology
514G3	Biological (monoclonal Ab)	*Staphylococcus aureus*	Phase II	NCT02357966	Xbiotech
AR-101 (Aerumab)	Biological (monoclonal Ab)	*Pseudomonas aeruginosa*	Phase II	NCT03027609	Aridis Pharmaceuticals
MEDI-3902 ^c^	Biological (monoclonal Ab)	*Pseudomonas aeruginosa*	Phase II	NCT02696902	AstraZeneca
MEDI-4893 (Suvratoxumab) ^c^	Biological (monoclonal Ab)	*Staphylococcus aureus*	Phase II	NCT02296320	AstraZeneca
LBP-EC01 ^d^	Biological (Bacteriophage)	*Escherichia coli*	Phase I	NCT04191148	Locus Biosciences
Phagebank ^d^	Biological (Bacteriophage)	*Escherichia coli* or *Klebsiella pneumoniae*	Phase I	NCT04287478	Adaptive Phage Therapeutics
DSTA-4637S	Biological (monoclonal Ab –drug conjugate)	*Staphylococcus aureus*	Phase I	NCT02596399, NCT03162250	Roche/Genetech

^a^*Staphylococcus aureus* is the only Gram-positive species, all other target species are Gram-negative. All target species are contained in the priority pathogens list released by the World Health Organization. ^b^ Small organic compounds with MW < 900 Da. ^c^ These monoclonal antibodies are being developed as preventative agents. ^d^ The Phase I development studies for these products were initiated after publication of the 2019 World Health Organization (WHO) clinical pipeline report.

**Table 2 antibiotics-09-00418-t002:** Narrow-spectrum agents in preclinical development ^a.^

Species	Number	Cellular Metabolism	Phage Products	Anti-Virulence	Direct Membrane	Cell Wall Synthesis	Immuno-Modulation	Not Disclosed
*Acinetobacter baumannii*	9	1	1		3		1	3
*Clostridium difficile*	8	3	1	2	1		1	
*Escherichia coli*	10		9	1				
*Helicobacter pylori*	2				2			
*Klebsiella pneumoniae*	1		1					
*Mycobacterium tuberculosis*	43	24		1	1	6	1	10
*Neisseria gonorrhoeae*	2	1						1
*Pseudomonas aeruginosa*	18	1	7	6	1	2	1	
*Staphylococcus aureus*	7		4	2				1
Total	100	30	23	12	8	8	4	15

^a^ Data adapted from WHO antibacterial preclinical pipeline review (17).

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
