# Peer review of "Narrow-Spectrum Antibacterial Agents—Benefits and Challenges"

_antibiotics, 2020, doi:10.3390/antibiotics9070418_

Round 1
Reviewer 1 Report
Comments for the authors -
Development of effective treatment strategy towards antimicrobial resistance to broad-spectrum antibiotics is currently the most important research goal to contain widespread infections across the world. The author’s comments on narrow-spectrum antibacterial agents – the benefits and challenges is well written and received.
Some considerations-
In the abstract and elsewhere the authors mention “non-traditional products” and discuss the other treatment options being explored. For example, 23 bacteriophage products and 12 anti-virulence approaches in the WHO list. Can Antimicrobial photodynamic therapy (aPDT) be considered as non-traditional technique or product? These methods are capable of inactivating pathogens efficiently without the risk of inducing resistances. This mode of treatment can be very effective for preventing hospital transmission and water remediation. Some examples of efficiently sparing the beneficial microbiome during irradiation and treatment especially in dentistry, skin and wound infections are described (Critical Reviews in Microbiology, 2018, 44:5, 571-589, DOI:10.1080/1040841X.2018.1467876).
Page 2/12 “However, there are other mechanisms that impact resistance emergence.” No citation is provided for this statement.
Additional information of gram negative or positive type of bacteria/pathogen can be included in table 1 to highlight the target identification of small molecule drugs and other approaches mentioned.
Author Response
Reviewer 1
In the abstract and elsewhere the authors mention “non-traditional products” and discuss the other treatment options being explored. For example, 23 bacteriophage products and 12 anti-virulence approaches in the WHO list. Can Antimicrobial photodynamic therapy (aPDT) be considered as non-traditional technique or product? These methods are capable of inactivating pathogens efficiently without the risk of inducing resistances. This mode of treatment can be very effective for preventing hospital transmission and water remediation. Some examples of efficiently sparing the beneficial microbiome during irradiation and treatment especially in dentistry, skin and wound infections are described (Critical Reviews in Microbiology, 2018, 44:5, 571-589, DOI:10.1080/1040841X.2018.1467876).
While antimicrobial photodynamic therapy can be considered as non-traditional, it is not narrow-spectrum which is the focus of this commentary and the Special Issue. For that reason, it is felt that it falls outside of the scope of this article.
Page 2/12 “However, there are other mechanisms that impact resistance emergence.” No citation is provided for this statement.
The reference to other mechanisms that impact resistance emergence beyond direct selection are the examples of resistance via transmission etc. that follow this statement. The first statement has been changed to add clarity.
Additional information of gram negative or positive type of bacteria/pathogen can be included in table 1 to highlight the target identification of small molecule drugs and other approaches mentioned.
The Gram-status of the species in Table 1 and their inclusion in the WHO’s priority pathogen list is included in Table 1
Reviewer 2 Report
In this commentary , the authors explored an interesting topic regarding the narrow spectrum antibacterial approach. Overall, the commentary is well written and detailed and explain the collateral damage that broad-spectrum antibacterial drugs have on the intestinal microbiome and the subsequent risks of adverse events. Furthermore, they specified the actual potentiality of current narrow spectrum.
The rising problem of MDR etiologies has, at the same time, led to a reduction in treatment options, shifting the focus on the search for new and effective preventive strategies, which could directly hit its pathogenesis. I suggest to add a short comment about the pathogenesis-targeted strategies to prevent MDR addressed by recent review (Cotoia et al, Microganism 2020). Please add this reference.
When report “Rapid initiation of appropriate treatment correlates with reduced mortality and improved clinical outcomes”, please add appropriate reference
Please clarify. What do you mean with “small molecules” reported in table 1.
I suggest to separate de description of table 1 and 2 in the main text
Author Response
Reviewer 2
The rising problem of MDR etiologies has, at the same time, led to a reduction in treatment options, shifting the focus on the search for new and effective preventive strategies, which could directly hit its pathogenesis. I suggest to add a short comment about the pathogenesis-targeted strategies to prevent MDR addressed by recent review (Cotoia et al, Microganism 2020). Please add this reference.
The reviewer is correct that some of the strategies listed in the above referenced article have shown promise in the reduction of infections. However, the strategies reviewed in this article, like oral chlorhexidine hygiene, chlorhexidine bathing, and universal gloving and contact isolation are aimed at non-selective decolonization and infection control strategies that do not qualify as targeted narrow-spectrum agents. Such infection control therapies are welcome additions to the fight against MDR infections and should be pursued, but unfortunately do not fall within the scope of this article
When report “Rapid initiation of appropriate treatment correlates with reduced mortality and improved clinical outcomes”, please add appropriate reference.
Some appropriate references added
Please clarify. What do you mean with “small molecules” reported in table 1.
Footnote with description added to Table 1
I suggest to separate de description of table 1 and 2 in the main text
Sentence reworded to include separate citation to Tables.